# OpenReview forum: "Agentic AI for Enhanced Research Paper Authoring: A System Featuring Verifiable Citation and Intelligent Data Integration NLDL #42"
_NLDL.org/2026/Conference — Submitted to NLDL 2026_

### Official Review · Reviewer_2S7y · 2025-09-16
**not a full paper**

**Rating:** 1
**Confidence:** 5
**Final Rating:** 1
**Final Confidence:** 5

**Summary:**

This paper proposes an agentic AI system for research authoring that emphasizes verifiable citation management, intelligent chart integration, and multilingual support to improve reliability, accessibility, and efficiency in academic writing.

**Strengths:**

Almost none.

**Weaknesses:**

No method, no experiment, no results. This paper does not meet the basic requirements of a full paper.

**Final Justification:**

There was no author response.

**Justification:**

No method, no experiment, no results. This paper does not meet the basic requirements of a full paper.

---

### Official Review · Reviewer_wmNV · 2025-09-18
**Reject - Lack of methodology and empirical testing.**

**Rating:** 1
**Confidence:** 5

**Summary:**

The paper proposes a system for AI-agent-driven development of research papers. The paper focuses on three key aspects of the system:
- Citation Accuracy - Here, the authors propose to link external research knowledge bases to support the citation process.
- Chart Interpretation - Here, the authors propose to integrate various technologies like OCR, Chart Detection, Trend Analysis, and Text Summarization to improve the system's multi-model capabilities.
- Multilingual Support - Here, the authors emphasize the importance of supporting non-English native speakers.
The paper further discusses the importance of well-working tools and frameworks for academic writing and reviewing, thus further addressing the challenges that their work attempts to overcome. The paper is quite short and does not include any detailed method or implementation, nor any empirical experiments that support the authors' claims related to Citation Accuracy, Chart Interpretation, or Multilingual Support features described in sections 2, 3, and 4.

**Strengths:**

The paper addresses an evergoing and important aspect of academic research, namely, how to ensure that scientific ideas and results are communicated clearly and accurately. The authors describe various challenges found within the academic sphere and how AI-based software solutions could be promising tools to mitigate these challenges. The paper also proposes some design aspects and architectural ideas that should be incorporated into any academic writing and reviewing tool.

The language is clear and well-written, and the paper is well-cited for the claimed challenges.

**Weaknesses:**

The article addresses an important problem statement and argues well for why there is a need for a solution, but there is a lack of methodology and empirical testing. The authors promote their implementation, named "42AI", which they claim overcomes the aforementioned challenges within academic writing and reviewing. Nevertheless, the implementation or design of 42AI is never shown, not written, nor illustrated. This is a must for such a paper to describe the implementation and how the design choices contribute to overcoming all the challenges laid out in the paper. A "Method" section should be included that describes the implementation, preferably with graphical illustrations.

Further, the paper lacks empirical testing to strengthen the author's claims related to 42AI. There should be tests and comparisons that showcase 42AI's performance within Citation Accuracy, Chart Interpretation, and Multilingual Support. Without such empirical data, the author's words lack support. Running 42AI on a benchmark like CiteBench would provide a better basis for comparing 42AI to alternative implementations.

As it stands, the paper discusses challenges in academic writing and reviewing, and how 42AI addresses these. However, after reading the paper, it is not clear how well 42AI addresses the challenges, nor what parts of 42AI's design contribute to its claimed strengths. The paper appears more like a marketing description of a new software than an academic article, as it says the "what" but not the "how". Please include an in-depth Method section and a Result section with empirical data that support the claims.

**Justification:**

The paper points out challenges in academic writing and reviewing, and claims its solution, named "42AI", overcomes these challenges. But they do not provide any insight into the design of 42AI, thus not sharing any gained knowledge on how to design such systems. Also, the authors do not include any empirical evidence for their claims that 42AI overcomes the aforementioned challenges; thus, the reader is left to trust the authors' word. The lack of these two vital components, namely, a strong Methodology section and a Result section, which would be expected for such a paper, is the core reason for the rejection.

---

### Official Review · Reviewer_9CLL · 2025-10-03
**Review of Agentic AI for Enhanced Research Paper Authoring: A System Featuring Verifiable Citation and Intelligent Data Integration NLDL #42**

**Rating:** 1
**Confidence:** 5
**Final Rating:** 1
**Final Confidence:** 5

**Summary:**

This paper proposes an agentic system for improving the research paper writing process along three axes.  The first axis is ensuring verifiable citations, the second axis is automatically analyzing chart-based data to create new written content in the paper, and the third axis is adding multilingual support to facilitate international, collaborative research.

**Strengths:**

- The overall prose and writing was clear, and easy to follow.
- AI research assistants are currently a ``hot" topic.

**Weaknesses:**

- This paper was only 2.5 pages in length, whereas the length of a full paper should be between 5-8 pages.  Perhaps this was intended as an extended abstract?  The content does not seem sufficient to form a research paper.
- It is unclear what the contributions of this paper are, as a research paper.
- It is unclear why the system they propose is novel.   Many frontier models are able to make sense of data, automatically detect charts, and extract data, and formulate correct citations.   With regards to multilingual support, it is unclear why their system would be more important than any state-of-the-art translation system.
- Many of the design choices are very vague. For instance: Lines 178-181: “Addressing this, our system incorporates features designed to mitigate such biases and ensure a more equitable user experience.”  There is no description of the features in more detail.  Another example:
Lines 192-194: “Furthermore, the system’s capabilities in semantic textual similarity [23] enhance its ability to accurately assess and synthesize information from multilingual sources.”   What is unique about the proposed system that would do better than the state-of-the-art solutions?
- The system is under-specified; it’s not clear why existing systems (frontier models) cannot accomplish what is described here.  There are much more sophisticated systems for AI research agents, such as Google’s AI Co-scientist.
- Without a prototype implementation, the benefits of the proposed system over existing frontier models remains unclear.

**Final Justification:**

There was no author response, and I stand by my initial review.

**Justification:**

While AI research assistants are a hot topic, there is not enough technical content here to be considered a research paper.  It reads more like an extended abstract. Furthermore, the content itself is not novel.  Specifically, it is unclear why existing frontier models would perform worse or cannot perform the capabilities described in this system (and there is no discussion surrounding this).  Based on the lack of novel technical content, I recommend a reject decision.

---

### Official Review · Reviewer_6VZ7 · 2025-10-10

**Rating:** 1
**Confidence:** 5
**Final Rating:** 1
**Final Confidence:** 5

**Summary:**

The authors describe a system that assists researchers in writing paper, for instance by suggesting references, translation, automatically sense-making of graphs/charts, etc.

**Strengths:**

The described system could be very helpful in daily work of researchers.

**Weaknesses:**

The system is rather advertised than described. There is not a single sentence providing details of how the system is assembled, evaluated or guaranteed to work as expected.

**Final Justification:**

There was no author response, see comments in the review.

**Justification:**

We always strive for understanding and learning. The writeup however does not disclose any detail and is not falsifiable in that respect. It is rather a promise than a scientific paper. By contrast, we want to understand how things work, how they are integrated, how it can be guaranteed that the output is correct etc. The writeup does not provide any of these answers.

---

### Meta-Review · Area_Chair_j2w8 · 2025-11-01

**Recommendation:** Reject
**Confidence:** 5

**Metareview:**

The paper addresses a common problem when using LLM: the attribution of work to the proper source.

The paper description is rather limited in length.

The content of the paper looks more like a specification of a desired system than a description and evaluation of an actual system.

While the problem tackled could be of interest to the conference community, the quality of the exposition is not appropriate.

Therefore, I would suggest rejecting the paper in its current shape.

---

### Decision · Program_Chairs · 2025-11-05

**Decision:**

Reject

**Comment:**

Based on the reviewers and AC comments, the paper cannot be presented at the conference.